# Changes in mean evapotranspiration dominate groundwater recharge in semi-arid regions

Tuvia Turkeltaub[1] and Golan Bel[2,3]

[1]Zuckerberg Institute for Water Research, Blaustein Institutes for Desert Research, Ben-Gurion University of the Negev, Sede Boqer Campus 8499000, Israel
[2]Department of Environmental Physics, Blaustein Institutes for Desert Research, Ben-Gurion University of the Negev, Sede Boqer Campus 8499000, Israel
[3]Faculty of Environmental Sciences, Czech University of Life Sciences Prague, Kamýcká 129, Praha – Suchdol, 165 00, Czechia

**Correspondence:** Tuvia Turkeltaub (tuviat@bgu.ac.il)

**Abstract.** Groundwater is one of the most essential natural resources and is affected by climate variability. However, our understanding of the effects of climate on groundwater recharge (R), particularly in dry regions, is limited. Future climate projections suggest changes in many statistical characteristics of the potential evapotranspiration ($E_p$) and the rainfall that dictates the R. To better understand the relationship between climate statistics and R, we separately considered changes to the mean, standard deviation, and extreme statistics of the $E_p$ and the precipitation (P). We simulated the R under different climate conditions in multiple semi-arid and arid locations worldwide. Obviously, lower precipitation is expected to result in lower groundwater recharge and vice versa. However, the relationship between R and P is non-linear. Examining the ratio R/P is useful for revealing the underlying relation between R and P; therefore, we focus on this ratio. We find that changes in the average $E_p$ have the most significant impact on R/P. Interestingly, we find that changes in the extreme $E_p$ statistics have much weaker effects on R/P than changes in extreme P statistics. Contradictory results of previous studies and predictions of future groundwater recharge may be explained by the differences in the projected climate statistics.

## 1 Introduction

Groundwater sustainability depends on balancing groundwater recharge (R) and groundwater abstraction (Hartmann et al., 2017; Wada et al., 2010; Collenteur et al., 2021; Singh et al., 2019; De Filippi and Sappa, 2024; Viaroli et al., 2018; Andualem et al., 2021). R is the amount of water infiltrating the soil deep enough such that it is not lost to evaporation, transpiration, or runoff. Note that this definition is not the same as the definition of some authors, which define it as the amount of water replenishing the aquifer (Healy and Scanlon, 2010) (the main difference is the travel time). Many areas across the globe show a growing dependence on groundwater resources, which will only increase in the future (Bierkens and Wada, 2019; Taylor et al., 2013a, b). Climate variability affects both the precipitation and the evapotranspiration statistics. Therefore, understanding the potential effects of these factors on R is of great importance. In order to improve groundwater resource management and

reduce negative human effects (Taylor et al., 2013a; Vivek et al., 2024; Pino-Vargas et al., 2023; Sappa et al., 2019; Quandt et al., 2023), the direct influences of climate variability on R must be quantified.

In recent years, much effort has been devoted to the analysis of the sensitivity of groundwater systems to climate change (Meixner et al., 2016; Pulido-Velazquez et al., 2015; Touhami et al., 2015; Fu et al., 2019; Döll and Fiedler, 2008; Tillman et al., 2016; Reinecke et al., 2021; Huang et al., 2023; Berghuijs, 2024; Berghuijs et al., 2024; Lorenzi et al., 2024; Langman et al., 2022). However, no conclusive generic outcomes can be drawn regarding the relationship between changes in climate conditions and the resulting changes in R rates (Al Atawneh et al., 2021; Green et al., 2011). The main source of uncertainties in future R is the uncertainties in climate predictions. It is unclear whether the climate variability is amplified or smoothed in the R response (Taylor et al., 2013a; Field et al., 2012; Reinecke et al., 2021; Moeck et al., 2016). Moreover, even the trend of the R response is uncertain (Smerdon, 2017).

Climate variability may also change the seasonal distribution of the precipitation (P) (Allan and Soden, 2008; Field et al., 2012). Increasing temperatures are expected to increase evapotranspiration ($E_a$) (Condon et al., 2020) while the increased $CO_2$ concentrations are expected to lower the $E_a$ (Cao et al., 2010), and the overall effect is uncertain (Barnett et al., 2008). The future R uncertainties are even more dominant in arid and semi-arid regions where the variability of the $E_a$ affects the threshold values for R (Cuthbert et al., 2019b). Different studies have reached contradictory conclusions regarding the effects of climate change on R in arid and semi-arid regions (Crosbie et al., 2013). Some studies found that the changes in R are greater than the changes in climate conditions (Ng et al., 2010; McKenna and Sala, 2018), while others found weak sensitivity of semi-arid regions to climate variability (Döll and Fiedler, 2008; Cuthbert et al., 2019a). Under future climate conditions, the precipitation and potential ET ($E_p$) statistics and, particularly, the frequency of extreme events (Myhre et al., 2019) may change. The effects of these extreme events may lead to an increase (Taylor et al., 2013b, a; Cuthbert et al., 2019b; Shamsudduha and Taylor, 2020; Goni et al., 2021) or a decrease (Cuthbert et al., 2019b) in R.

Previous studies investigated the R response to predicted future climate conditions using global climate model (GCM) (Crosbie et al., 2013; Tillman et al., 2016) or regional GCM (Pulido-Velazquez et al., 2015) predictions. Pulido-Velazquez et al. (2015) also considered modifications of the mean and standard deviation (STD) of the P in the regional GCM predictions. However, these studies could not provide a conclusive understanding of the effects; in particular, changes in the $E_p$ statistics were not directly considered.

The main objective of this study is to explore the changes in diffuse (rather than focused, agricultural, or mountain, see Meixner et al. (2016)) R in semi-arid and arid regions due to changes in P and $E_p$ statistics. In areas where R occurs predominantly through focused processes, additional factors dominate the overall recharge and are beyond the scope of this study. Here, R is defined as the accumulated water flux at a 5 m depth, assuming that this flux reaches the water table. To enhance our understanding of the R response to climate variability, we separately consider the changes in the (1) mean, (2) STD, and (3) extreme events of $E_p$ and P statistics (relative to the measured statistics) in multiple locations across the globe. While we do not consider specific future climate projections, we identify and quantify the effects of changes in climate variables statistics on R in semi-arid and arid regions.

## 2  Methods

Groundwater recharge is a fraction of precipitation. Within the linear response regime, changes in precipitation would lead to a change in the recharge but not a change in the ratio between the groundwater recharge and the precipitation. To emphasize the nonlinear response, we present the ratio between the recharge and the precipitation and the changes in these ratios. We use a numerical model to simulate the R under atmospheric boundary conditions. In what follows, we provide the details of the model, the independent data used for verification of the model, and the changes applied to the climate variables statistics.

### 2.1  Groundwater recharge data and modeling

To explore the impact of climate statistics on R, we identified 196 semi-arid and arid locations (supporting information (SI)) in which R was estimated using ground-based methods such as chloride mass balance, water isotopes, etc. (Taylor et al., 2013b; Scanlon et al., 2006; Moeck et al., 2020). These locations span both hemispheres and different continents (see Fig. 1a). Furthermore, they cover a wide range of soil types (Fig. 1b) and climate conditions, including various seasonal P distributions relative to temperature and other factors affecting evapotranspiration. The locations are characterized by bare soil or sparse vegetation, where transpiration is negligible relative to the evaporation. For locations where the model did not perform well, factors such as land use change may explain the discrepancies between simulated and reported R. In some locations, over the last decades, R may have changed substantially due to various human modifications of landscapes, such as changes in land use and land cover, as well as water conservation works (Turkeltaub et al., 2018; Guillaume et al., 2009; B. et al., 1990). Since the R fluxes estimated by the ground-based methods represent an integration over varying timescales, they are likely to reflect different stages of this evolution.

Diffuse R fluxes were simulated using unsaturated flow modeling for these locations by numerically solving the 1D vertical Richards equation:

$$\frac{\partial \theta}{\partial t} = \frac{\partial}{\partial z}\left[K(\psi)\left(\frac{\partial \psi}{\partial z}+1\right)\right], \tag{1}$$

where $\psi$ is the matric potential head $[L]$, $\theta$ is the volumetric water content (dimensionless), $t$ is time $[T]$, $z$ is the vertical coordinate $[L]$, and $K(\psi)$ $[L\,T^{-1}]$ is the unsaturated hydraulic conductivity function. The Richards equation was numerically integrated using the Hydrus 1D (Šimůnek et al., 2009). Knowledge regarding the soil hydraulic functions is essential in order to solve the Richards equation. The soil retention curves and the unsaturated hydraulic curves are commonly described according to the van Genuchten-Mualem (VGM) model (Mualem, 1976; Van Genuchten, 1980):

$$S_e = \frac{\theta - \theta_r}{\theta_s - \theta_r} = \left[1+(\alpha|\psi|)^n\right]^{-m}, \tag{2}$$

where $S_e$ is the degree of saturation ($0 < S_e < 1$), $\theta_s$ and $\theta_r$ are the saturated and residual volumetric soil water contents,

respectively, and $\alpha$ $[L^{-1}]$, $n$, and $m = (1-1/n)$ are shape parameters. Hydraulic conductivity is assumed to behave according to:

$$K(S_e) = K_s S_e^l \left[ 1 - \left[ 1 - S_e^{1/m} \right]^m \right]^2 \tag{3}$$

where $K_s$ $[L\ T^{-1}]$ is the saturated hydraulic conductivity, and $l$ is the pore connectivity parameter prescribed as $0.5$.

Assuming that the unsaturated zone mainly consists of siliciclastic materials, the VGM parameters were determined using the pedo-transfer function ROSETTA (Zhang and Schaap, 2017), which uses a neural network to estimate the soil hydraulic parameters from soil attributes, such as soil texture and bulk density. Sand, silt, and clay contents and the bulk density were extracted at the considered locations from global soil maps reported by Hengl et al. (2014). Note that the data is divided into seven layers, but for the current study, only information from the top three layers was used (0–5, 5–15, and 15–30 cm; the VGM parameters are provided in SI). We assume that evaporation is mostly limited to the topsoil (Or et al., 2013); therefore, we only considered the heterogeneity of these levels. Furthermore, the water table depths at the investigated locations, which were extracted from the global map presented by Fan et al. (2013), indicated that in most locations, groundwater is below 0.8 m depth and no phreatic evaporation is expected (SI; Chengcheng et al., 2020, Hellwig, 1973).

The water flow simulations were carried out using atmospheric boundary conditions with surface runoff. Daily precipitation and $E_p$ (potential ET) values were specified at the upper boundary. The minimum allowed pressure head at the soil surface was constant (hCritA= 100000 cm). Lower boundary conditions were prescribed as free drainage, where the water flux across this boundary is considered the R. The depth of the simulated soil column was 500 cm, and it was discretized into 101 grid cells. A finer node spacing was implemented at the upper boundary, where the top node was three times thinner than the bottom node. Water content at field capacity was prescribed as the initial condition at the start of each simulation. Each simulation was run for 146,100 days, and the calculated daily R fluxes between days 73,050 and 146,100 were used for the analyses to avoid the influence of the initial conditions. In all the locations considered, the differences between the estimated and the reported R/P ratios were below 5%, illustrating the suitability of the model.

## 2.2 Climate data and generation of P and $E_p$ time series

The CRU TS 3.2 climate dataset (Harris et al., 2014), including daily values of precipitation and potential evapotranspiration ($E_p$), was used for the analyses presented in the current study. The datasets were temporally downscaled, following van Beek (2008), to daily values using ERA40 (1958–1978, Uppala et al. (2005) and ERA-Interim (1979–2015, Dee et al. (2011)). The daily $E_p$ values were calculated according to the Penman-Monteith method using climate variables such as mean, maximum, and minimum temperatures, vapor pressure, cloud cover, and wind speed (Harris et al., 2014). Stochastic P and $E_p$ time series spanning 400 years (146100 days) were synthesized based on these 58-year-long CRU TS 3.2 records of daily precipitation and potential evapotranspiration ($E_p$). P time series were established based on the empirical histograms of the number of rainy days and the daily P amount distributions (SI; Turkeltaub and Bel, 2023). The $E_p$ time series were established by random sampling

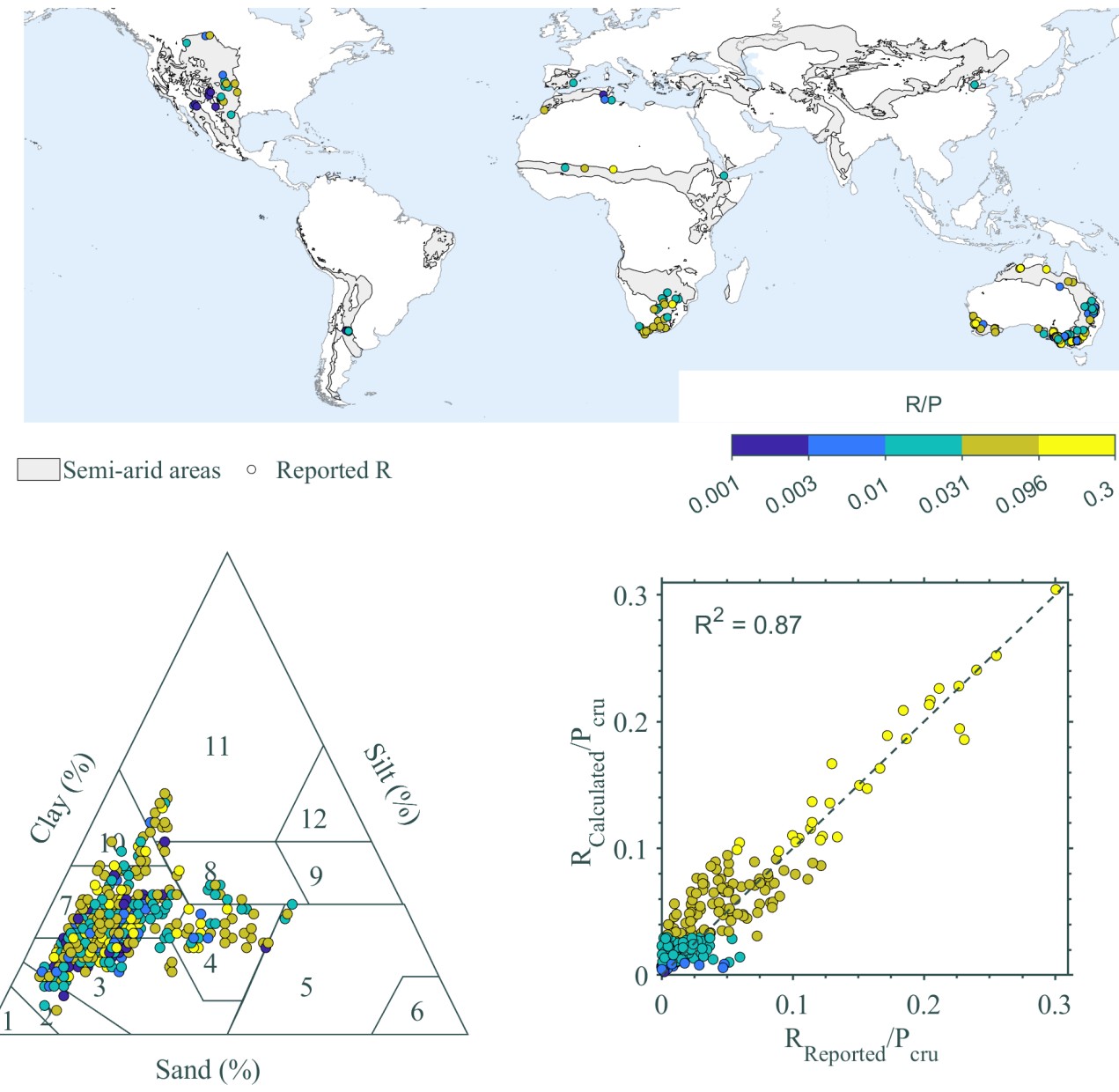

**Figure 1.** (a) The geographic distribution of the locations considered. (b) The soil composition, in terms of sand, silt, and clay, for all the locations considered in this study. The data are based on Hengl et al. (2014) and represent the reported soil characteristics for 0–5, 5–15, and 15–30 cm depths. (c) The simulated and reported ratio between the precipitation and the groundwater recharge for all the locations (In addition to the $R^2$ provided in the plot, other statistical indices for model performance evaluation are also provided: Mean Error = -0.0058; Mean Absolute Error = 0.016; Root Mean Square Error = 0.02.).

of $E_p$ empirical distributions for each calendar month separately (Turkeltaub and Bel, 2023). Furthermore, it was shown by the authors (Turkeltaub and Bel, 2023) that the best synthesis method involves a correction of the synthesized climate to match the observed monthly statistics. Thus, the P and $E_p$ time series were corrected accordingly (see the examples in the SI Figs. S1 and S2 for the effects of the correction on the monthly P and $E_p$ statistics).

## 2.3   Modification of the mean ($\mu$) and the standard deviation (STD, $\sigma$)

To examine the effects of changes in the statistics of the P and the $E_p$ time series on groundwater recharge (R), the yearly mean, $\mu$, and STD, $\sigma$, of these series were modified. Note that when we modify the average of a time series, the STD is conserved and vice versa. The modification of $\mu$ is simply conducted by adding to each value (in the P series, only to non-zero values) in the original time series the difference between the original and modified yearly average divided by the number of relevant days in that year. Note that this correction could possibly have resulted in negative daily P and $E_p$ amounts when considering

a reduction of the yearly averages. Therefore, for the $E_p$, only days with amounts above the correction were modified in order to ensure non-negative values of $E_p$ for all days. For the P series, we further wanted to conserve the statistics of the number of rainy days. Therefore, the maximal allowed reduction of P sets the value to 1 mm/day (Turkeltaub and Bel, 2023). The modification of the STD is done in two stages. Firstly, each value in the original time series is multiplied by the ratio between the original STD, $\sigma_{org}$, of the total yearly P or the $E_p$ and the modified STD, $\sigma_{mf}$. Subsequently, the differences between the

averages of the original and the modified time series are corrected according to the procedure described above to preserve the original mean. Mathematically, the correction method for the annual $\mu$ of a time series is described as:

$$\text{MTS}_\mu(t) = \begin{cases} \text{OTS}(t) \geq \text{Threshold} & \text{OTS}(t) + \frac{\Delta}{\text{Na(t)}} \\ \\ \text{OTS}(t) < \text{Threshold} & \text{OTS}(t) \end{cases} \tag{4}$$

where $\text{MTS}_\mu$ and OTS are the modified mean and the original time series, respectively. $\Delta$ is the difference between the modified and the original annual P or $E_p$ averages ($\Delta \equiv \langle \text{MTS}_\mu \rangle_a - \langle \text{OTS} \rangle_a$, and $\langle . \rangle_a$ is the annual average of the variable

represented in the time series). $\text{Na(t)}$ is the number of days in the year of time $t$, with P or $E_p$ values larger than the threshold. The threshold is defined as $-\Delta/\text{Na(t)}$ for the $E_p$ and as $-(\Delta/\text{Na(t)} + 1)$ for the P. $\text{Na(t)}$ is found recursively.

    To modify the $\sigma$ of a time series, the following transformation is applied:

$$\text{MTS}_\sigma(t) = \begin{cases} \text{OTS}(t) \times \frac{\sigma_m}{\sigma_o} \geq \text{Threshold} & \text{OTS}(t) \times \frac{\sigma_m}{\sigma_o} + \frac{\Delta_\sigma}{\text{Na(t)}} \\ \\ \text{OTS}(t) \times \frac{\sigma_m}{\sigma_o} < \text{Threshold} & \text{OTS}(t) \times \frac{\sigma_m}{\sigma_o} \end{cases} \tag{5}$$

where $\sigma_m$ and $\sigma_o$ are the modified and original yearly standard deviation of the time series, respectively. $\Delta_\sigma \equiv \langle \text{OTS} \rangle_a \left( \frac{\sigma_m}{\sigma_o} - 1 \right)$.

The threshold is defined similarly to the definition above for the mean modification. Fig. 2 depicts an example of the modification of the average, $\mu$, and STD, $\sigma$, of the yearly P for a specific location ([-36.4469, 145.711]; (Crosbie et al., 2010)).

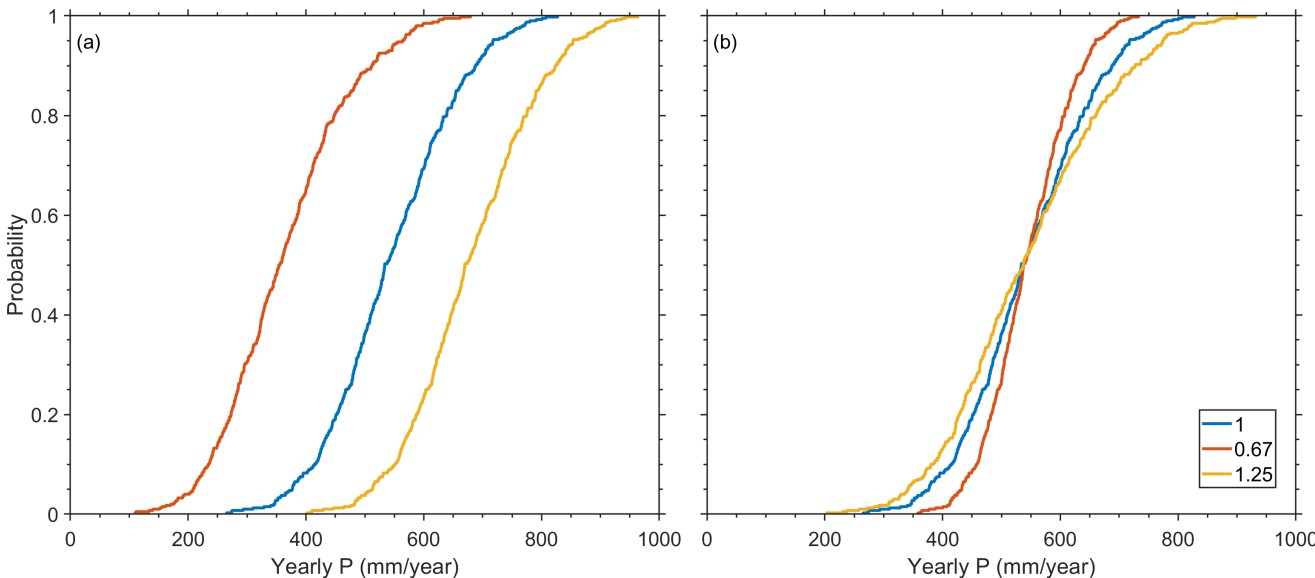

**Figure 2.** An example for the modification of the (a) mean, $\mu$, and (b) the STD, $\sigma$, of a P time series for a specific location where a groundwater recharge flux was reported ([-36.4469, 145.711]; (Crosbie et al., 2010)). The different colors correspond to the indicated modification factors.

## 2.4 Modification of extreme events statistics

In order to increase the frequency of extreme events, the time series were modified such that the mean was conserved, and events

above a specified quantile (0.98, 0.95, or 0.9) were doubled (Myhre et al., 2019; Fischer and Knutti, 2016). The doubling was done by randomly selecting events with values below the specified quantile and replacing their original value with a value from those above the quantile. We established two separate scenarios. The first doubles the frequency of the extreme events without preserving the seasonal cycle. Namely, an extreme event may be introduced into any day in the original series regardless of the season. In the second scenario, we preserve the seasonal cycle and double the extreme events for each calendar month

separately. In the latter case, the added extreme events correspond to observed events in the same calendar month. For both scenarios, in order to preserve the annual mean, we used a procedure similar to the one outlined above for the annual mean modification (see eq. (4)).

For the P time series, this was done without modifying the statistics of the number of rainy days; namely, only rainy days in the original time series could be randomly selected to receive one of the doubled extreme values. For the $E_p$ time series, there

was no additional constraint.

## 3 Results

### 3.1 Effects of changes in the mean

The first change we considered is a simple change in the mean ($\mu$) of climatic variables (P and $E_p$). Changes to the average $E_p$ affect the R/P ratio in all the locations (see Fig. 3). A larger mean $E_p$ ($\mu_m/\mu_0 > 1$) reduces the R/P ratio, while a smaller mean $E_p$ ($\mu_m/\mu_0 < 1$) increases the ratio. The result is expected because the larger the $E_a$ (i.e., the estimated amount of water evaporated, determined by the numerical model; $E_p \geq E_a$), the smaller the ratio (in this case, the P amount does not change). However, the change in the ratio is not the same for all locations, and obviously, it depends on the amount of water available for the actual $E_a$ and the amount of P. It also suggests a nonlinear relation between the R and P rates. Histograms of the distribution of the change in the ratio, $\left(\frac{R}{P}\right)_o - \left(\frac{R}{P}\right)_m$ (the subscripts $0$ and $m$ correspond to the original and the modified $E_p$ statistics, respectively), in the different locations, are depicted in Figure 4a-d. Some changes in the R/P ratio are expressed by an increase or decrease of up to 20% in the fraction of rainfall that becomes R (Figure 4a-d). Note that the largest change in R/P ratio occurs for a reduction of 0.67 of the mean annual $E_p$ (Figure 4a).

Similarly, we considered changes in the mean P (Fig. S3). The annual distribution of the P is not modified, and only the amounts are increased or reduced by the desired multiplicative factor (see the Methods section for details; see also Fig. 2). In Fig. S3, the ratios, R/P, under different changes to the mean P, are shown. In general, reducing the P results in a higher fraction of locations with a smaller R/P ratio, while increasing the mean P results in a higher fraction of locations with larger ratios. The histograms of the fraction of locations with different changes in the ratio (Figure 4e-h) reveal a more interesting response. Lowering the mean P results in a mixed response. The R/P ratio decreases in some locations, while it increases in others. In most locations with summer P (92%, 73 of the considered locations), decreasing the mean annual P results in an increase in the ratio (Figure 4e,f). The R in these locations is mostly a result of large P events, and the decrease in the mean P hardly affects the fraction of R during these events.

### 3.2 Effects of changes in the STD

The second change in the statistics of the climate variables that we considered is a change to the STD of the variables while keeping the mean unchanged (see Fig. 2). This is equivalent to uniformly broadening the distribution of the $E_p$ or the P (see the Methods section for details; Fig. 2). We find that increasing the STD of the $E_p$ (Figs. S4) or the P (Figs. S5) results in increasing the R/P ratio in almost all the locations. This is also reflected by the change in the ratio, where most estimated R/P ratios are larger than the original ratios, as indicated by the negative differences (Figure 4k,l,o,p).

Reducing the $E_p$ (Figs. S4) or the P (Figs. S5) STD reduces the R/P ratio in most locations. This is illustrated by the positive values of the R/P differences (Figure 4i,j,m,n). Overall, the reduction in the STD of the $E_p$ or the P has the smallest effect on the R/P ratio.

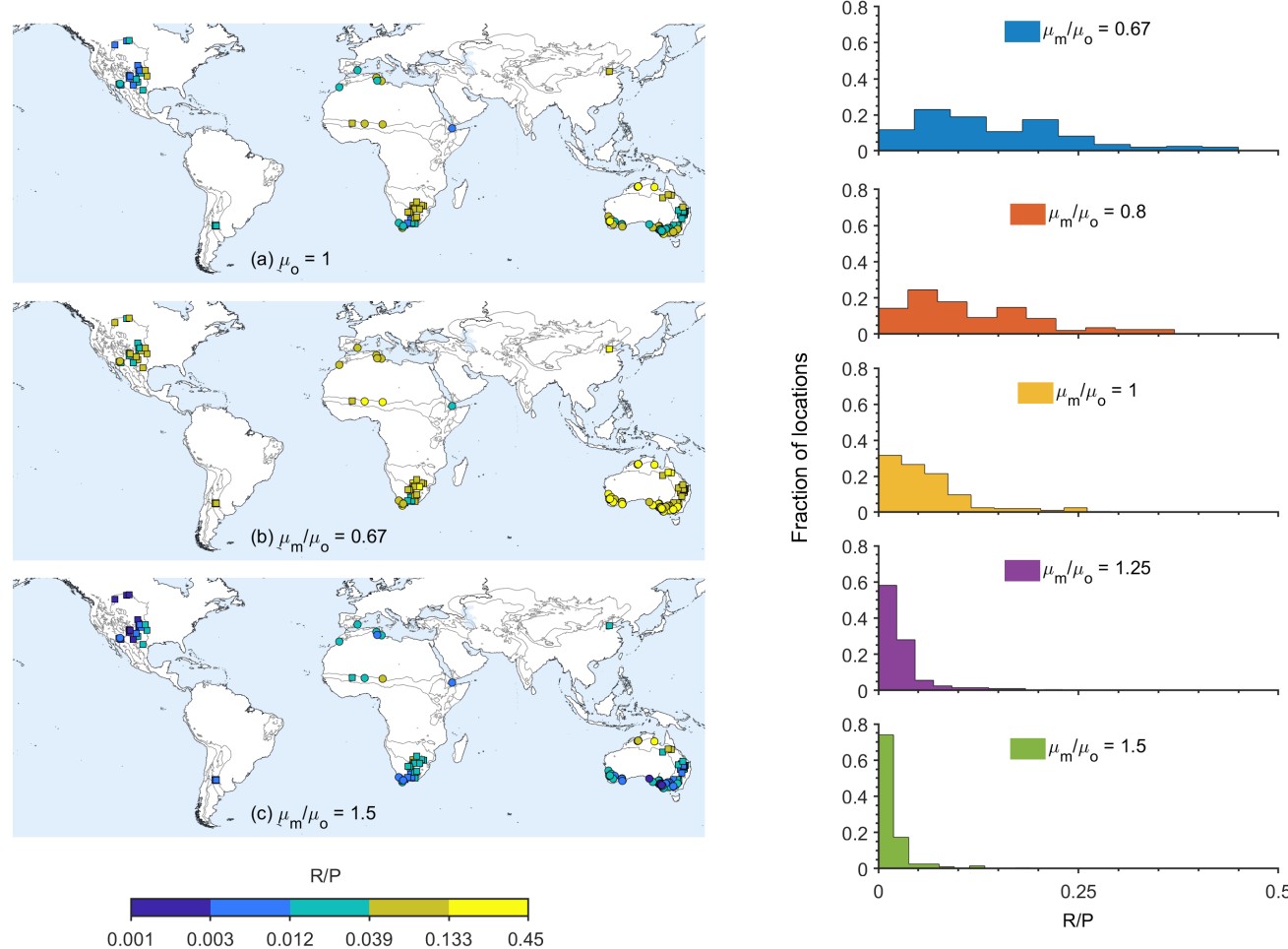

**Figure 3.** The effect of $E_p$ yearly mean ($\mu$) modification on the R/P ratio. The left panels depict the modified ratio in different locations for the measured (top panel), reduced by a factor of 2/3 (middle panel) and increased by a factor of 1.5 (bottom panel), mean annual $E_p$, respectively. The right panels depict the histograms of the fraction of locations with different R/P ratios under different mean $E_p$ modifications. The ratios between the modified annual mean $E_p$ ($\mu_m$) and the original one ($\mu_0$) are denoted in the figure.

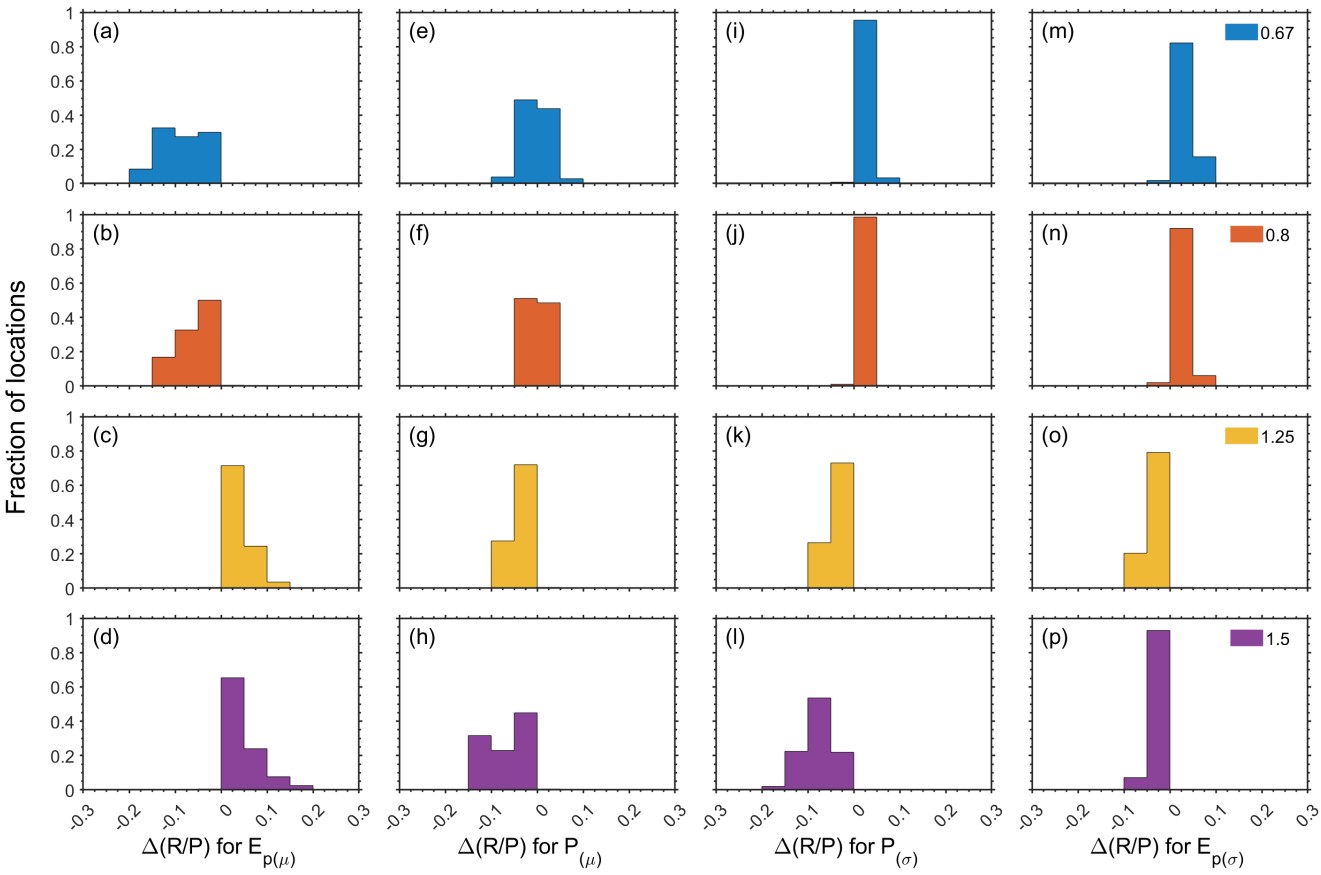

**Figure 4.** The panels depict the histograms of the change in the ratio R/P ($\Delta(R/P) = \left(\frac{R}{P}\right)_o - \left(\frac{R}{P}\right)_m$; the subscripts $o$ and $m$ correspond to the original and the modified statistics, respectively) for changes in the (a-d) $E_p$ yearly mean, (e-h) P yearly mean, (i-l) P yearly STD and (m-p) $E_p$ yearly STD. The colors indicate the modification factors, which are the ratios between the modified annual mean ($\mu_m$) or STD ($\sigma_m$) and the original one ($\mu_o$) or STD ($\sigma_o$).

### 3.3 Effects of changes in the frequency of extreme events

Under some future climate predictions, the frequency of extreme events is expected to double (Myhre et al., 2019). Therefore, modifications of the extreme statistics as outlined in subsection 2.4 were considered. In Fig. 5, the histograms of the fraction of locations with specific R/P ratios are depicted for the doubling of P (panels (a) and (c)) and $E_p$ (panels (b) and (d)) extreme
195 events above the 90%, 95%, and 98% quantiles. For reference, the panels also depict the histogram based on the measured climate conditions. It is apparent in panel (a) that doubling the extreme P events results in more locations with higher R/P ratios. Note that in this change, the total P is not modified; therefore, the increase in the ratio implies an increase in the actual R. The results are similar in panel (c) where the extreme events of each calendar month were doubled. In Fig. S6, the differences in R/P are presented showing that for all the locations considered, increasing the frequency of extreme events increases the R

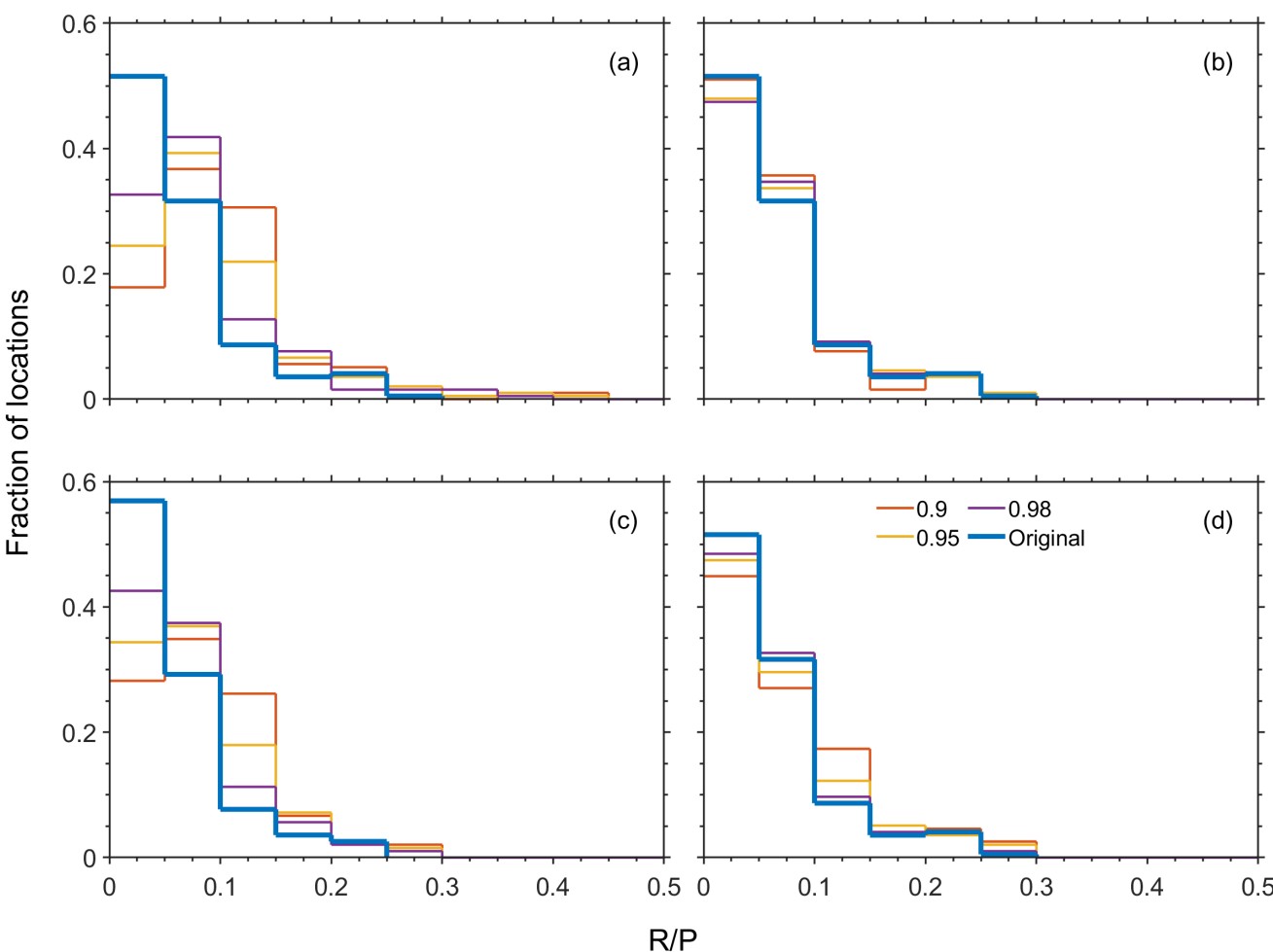

**Figure 5.** The fraction of locations with the specified R/P ratios due to an increase in the occurrence of annual extreme (a) daily P, or (b) daily $E_p$. The ratios due to an increase in the occurrence of extreme P or $E_p$ events for each calendar month separately are depicted in panels (c) and (d), respectively. See the Methods section for a detailed explanation of the changes to the extreme events statistics. The values of 0.98, 0.95, or 0.9 represent specified quantiles, where all events above the corresponding values were doubled to establish extreme climate scenarios. The blue line ('original') corresponds to scenarios with no changes to the extreme statistics.

200 despite the fact that the total P is unchanged. Panel (b) shows that doubling the extreme $E_p$ has a much smaller effect on the R. Preserving the seasonality while doubling the extreme $E_p$ events results in a somewhat stronger effect and more locations with higher R as shown in panel (d) and Fig. S6. In panels (b) and (d) of Fig. S6, it is shown that in most locations, doubling the extreme $E_p$ results in larger R, while in a small fraction of the locations, it reduces the R. Note that these results differ from those of increasing the $E_p$ STD for which all locations showed an increase in the R.

 **4 Discussion**

Understanding the response of R to varying climate conditions involves a broad range of possible changes to the statistics of the climate variables and renders the task overly complicated. Here, we investigated the effects of various aspects of the climate statistics on groundwater recharge in semi-arid and arid locations. We used Richards' equation (with the VGM hydraulic functions) to simulate the groundwater recharge under varying climate conditions. The use of the Richards' equation assumes that groundwater recharge is dominated by diffuse recharge. In regions with significant topography and rocky terrains, considerable runoff is expected (Casenave and Valentin, 1992; Mounirou et al., 2021). Furthermore, in regions where focused recharge and preferential flow paths prevail (van Schaik et al., 2008), using Richards' equation might not accurately capture the dynamics of groundwater recharge (Mirus and Nimmo, 2013; Hartmann et al., 2015, 2017; Appels et al., 2015). Therefore, we limit our analysis to regions where independent estimation of R agreed with our simulated R, suggesting that our method is adequate for these locations. In addition, as expected from the agreement, we found that runoff was not dominant in our simulations.

In our analyses above, we attempted to deal with this complexity by separately considering different changes in climate statistics. Reducing the mean P resulted in mixed outcomes. Some locations illustrated a decrease in the R/P ratio, while in others, it increased. The increase in the R/P ratio mostly occurred in locations with summer P. Two main explanations are suggested for this counter-intuitive change. The first reason is related to the P characteristics in such regions of heavy P events that promote deep drainage and R. The decrease in the mean P hardly affects the fraction of R during these events. Therefore, the ratio is increased because the R only decreases slightly while the P substantially decreases. An additional explanation is associated with reducing the amount of water available for evaporation. The reduction of the amount of P results in longer periods over which the $E_a$ is smaller than the $E_p$, thereby increasing the fraction of P going to the R (in most days, the $E_a$ is either equal to the $E_p$ when there is a continuum of water reaching the topsoil, or close to zero when there is not). Only 25% of the locations with winter P showed the same behavior–most likely because the $E_p$ during the rainy season is relatively low in these locations, and the effect of the second process mentioned above is weaker. When the mean annual P increases, all locations show a larger R/P ratio (see Fig.4e-h and Fig. S5).

We find that the R/P ratio shows higher sensitivity to changes in the mean $E_p$ than to changes in the mean annual P. The P is equal to the sum of the R and $E_a$ under the assumption that in a steady state, the change in the total soil water storage is negligible and the assumption that the runoff is negligible (this was verified for the locations considered in this study). Mathematically, we express it as:

$$R = P - E_a,$$

and, therefore,

$$\frac{R}{P} = 1 - \frac{E_a}{P}.$$

(6)

(7)

If one assumes that the evapotranspiration is a function of the ratio $E_p/P$ (Budyko, 1974), the changes in R/P are expected to be the same regardless of whether the change in $E_a$ is due to changes in $E_p$ or changes in P. However, if $E_a$ depends not only on $E_p/P$ but also on the actual water content of the top soil layer (Gerrits et al., 2009), one can expect different sensitivity to changes in $E_p$ or changes in P, as we observed.

The response to changes in the P STD is easily understood considering the fact that most of the R is triggered by large P events. The increase in the R/P ratio due to an increase in the $E_p$ STD can be attributed to the fact that in some years, the $E_p$ becomes small enough to allow significant R, while during the years with higher $E_p$, the reduction in R is much smaller because there is not always water available for $E_a$, i.e., larger values of $E_p$ do not affect the $E_a$ because it has already reached an upper limit. Ultimately, we found that changes in the statistics of extreme rainfall events have a much greater effect on R than in extreme $E_p$ events.

## 5 Conclusions

Understanding the combined effects of all the changes in climate variables on groundwater recharge is an ongoing effort and is expected to play a critical role in future studies. Many factors affect groundwater recharge, making it a complicated process to quantify. Some of these processes are hard to predict and others are related to human activity affecting directly (such as urbanization) and indirectly (such as deforestation) the fraction of precipitation infiltrating deep soil levels. Anthropogenic factors, such as land use changes, are expected to strongly affect groundwater recharge. Those changes are expected to occur much faster than changes in climate statistics. In our study, we examined the effects of changes in climate statistics, i.e., non-anthropogenic, on groundwater recharge in arid and semi-arid regions.

We considered locations worldwide. The selected locations are characterized by bare soil or sparse vegetation to avoid the effect of water loss due to transpiration. Furthermore, the site selection process included comparisons of simulated and reported yearly groundwater recharge fluxes to verify that only sites for which the model represents the natural conditions and considered and locations influenced by factors such as human disturbances are excluded. Despite the simplicity of the modeling approach, we found that in many places, worldwide, the model provides good estimates of the fraction of precipitation infiltrating deep soil.

Our simulations show that changes in climate statistics may have various effects on groundwater recharge. In most locations, increasing the mean P results in higher R/P ratios and vice versa while increasing the mean $E_p$ reduces the R/P ratio and vice versa. Increasing the STD of both P and $E_p$ results in a higher R/P ratio. In most locations doubling the frequency of extreme P or $E_p$ events results in a higher R/P ratio. However, the effect of more frequent P events is stronger than the effect of more frequent $E_p$ events.

Previous studies suggested different trends and changes in future groundwater recharge. The differences between the projected climate statistics used in those studies may explain those seemingly contradictory assessments of future groundwater recharge fluxes. To enhance our understanding and better explain the predicted groundwater recharge changes, results should be augmented by an analysis of the projected changes in potential evapotranspiration and rainfall statistics. As demonstrated

above, considering only changes in the mean and the STD is not enough, and changes in the statistics of extreme events are essential. This is attributed to the nonlinear responses of groundwater recharge to changes in climate statistics. The conclusions drawn in this analysis are valid for locations where diffuse recharge dominates the overall recharge. In areas where groundwater recharge occurs predominantly through focused processes (e.g., preferential flow and recharge of runoff at specific locations on the landscape), future analyses should include additional factors at the sub-catchment scale, such as topographic attributes and spatiotemporal variability in precipitation. Separate studies are required to investigate the effects of climate change on groundwater recharge in humid regions or under agricultural fields, where the root uptake and the transpiration are significant.

*Code and data availability.* The data and code used in this research are available at the following link:
http://www.hydroshare.org/resource/5a28b74c55c74a7e9ad2b59a0a5d9ab3

*Author contributions.* TT and GB designed the research, analyzed the data, and wrote the manuscript. TT performed the numerical simulations.

*Competing interests.* The authors declare no competing interests.

*Acknowledgements.* G. B. acknowledges support from, AdAgriF - Advanced methods of greenhouse gases emission reduction and sequestration in agriculture and forest landscape for climate change mitigation ($CZ.02.01.01/00/22_008/0004635$), and Israel Science Foundation (ISF), Grant No. 547/23. G. B. thanks Ester Levi for a lifelong inspiration.

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
