# Peer review of "Changes in mean evapotranspiration dominate groundwater recharge in semi-arid regions"

_EGUsphere, 2024_

## Referee Comment (RC2)

[referee-annotated manuscript omitted]

---

## Author Comment (AC1)

**Response to comments made by Reviewer 1 on a manuscript entitled 'Changes in mean evapotranspiration dominate groundwater recharge in semi-arid regions'**

Tuvia Turkeltaub[1] and Golan Bel[2]

[1]Zuckerberg Institute for Water Research, Blaustein Institutes for Desert Research, Ben-Gurion University of the Negev, Sede Boqer Campus 8499000, Israel
[2]Department of Environmental Physics, Blaustein Institutes for Desert Research, Ben-Gurion University of the Negev, Sede Boqer Campus 8499000, Israel

**Correspondence:** Tuvia Turkeltaub (tuviat@bgu.ac.il)

**General comment 1:** A major conclusion seems to misrepresent actual recharge changes. Namely, it is concluded that mean ET has a bigger influence on recharge than P, but the latter is only true when the ratio of recharge to precipitation is considered, but absolute (in mm/y) or relatively (in %) changes in recharge are very likely much bigger due to precipitation changes. Such an amplifying effect of precipitation on recharge (versus PET on recharge) is because changes in precipitation both the ratio of this precipitation becoming recharge, and the total amount of precipitation that can become recharge. In contrast, changes in PET only affect the ratio of precipitation becoming recharge. The latter is also highlighted in cited most recent work on the climate sensitivity of recharge.

**Reply to general comment 1:** *Indeed, our choice to focus on changes in the ratio highlights the nonlinear effects rather than the actual changes in the recharge (R). Actual changes in the recharge are meaningless without relating them to the changes in the precipitation (P). Even the simplest linear models assume that*

$$R = cP$$

*would predict a change in the actual recharge when the precipitation changes. However, when the relation is nonlinear, it is not clear what would be the response to different changes in P, and this is the point we wish to emphasize. To provide the requested information, we added to the SI figures showing the actual recharge as requested.*

*In this study, we do not attempt to predict the changes to the precipitation and the potential evapotranspiration ($E_p$). Rather, we focus on modeling the response of the groundwater recharge to assigned changes in the P and the $E_p$ statistics.*

*We added an explanation of this point at the beginning of the Methods section of the revised MS. We added figures presenting the actual recharge fluxes.*

**General comment 2:** The model claims to be accurate within 5% of recharge but given that recharge/rainfall is typically very low in these arid places so being within 5% can still mean the recharge is off by a lot (for example several 100%). These

uncertainties are not reflected in the projections. In addition, several data points appear to exceed the 5% error? More generally it is unclear why the climate projections can be considered accurate?

**Reply to general comment 2:** *We thank the reviewer for the comment. We double-checked our code and found that due to a typo, 7 locations in Africa showed an error that was larger than 0.05 in the ratio. Therefore, we corrected the code and redid the entire analysis. The corrections resulted in the removal of the 7 locations and the addition of 3 other locations in Africa that were within the 5% error.*

*As for the second part of the comment, indeed, for locations where the ratio between the recharge and the precipitation is small, the relative error (associated with absolute errors in the ratio of 0.05) may be very large. However, in most locations with reported small R/P ratios the uncertainty of the measurements is very large. To illustrate this issue, we show in the figure below the measured ratios in the locations in Africa together with the associated uncertainties Taylor et al. (2013). As shown in the figure below, an error of 0.05 in the R/P ratio is still within the reported uncertainty range for these locations. While we do not have the data regarding the uncertainties in all locations considered, we believe that the uncertainties show similar patterns.*

*We never claimed that climate projections are accurate, nor have we used them. The CRU data is not based on climate projections but, rather, on reanalysis data. The latter is effectively an interpolation of meteorological measurements.*

[Figure]

**Figure 1.** A comparison between reported GR/Rain ratio ranges and simulated ratios. Note that the name of each site used in our study, as reported by Taylor et al. (2013), is designated in the x-axis tick labels.

**General comment 3:** The mathematical derivation (Eq. 6-9) is applicable for ET but not for ETref. The argument that is made (that ET is proportional to the ETref) to circumvent this problem is not valid in semi-arid systems. If we follow the Budyko framework as a reference, one can see that in extremely energy-limited systems indeed ET is expected to (almost exactly) linearly (and one-on-one) scale with ETref. However, in more arid places changes in ETref will not be associated will similar ET changes, nor will be their relationship be linear.The expectation of how this behaves can be exemplified using simplified

version of the Budyko framework: E/P = 1-exp(-$\phi$) = E/P= 1-exp(-Eref/P). Thus: E= P*(1-exp(-ETref /P)). Thus, dE/d(ETref )= P/(P - exp(ETref/P) P + ETref), which is a nonlinear function for ETref > P. Therefore, the physical relevance of the derivation provided in the manuscript remains unclear to me.

40 **Reply to general comment 3:** *We thank the reviewer for this helpful comment. Indeed, we find that in many locations there is no linear relationship between the actual evapotranspiration ($E_a$) and the potential evapotranspiration ($E_p$). However, we also found that the Budoyko formula does not explain the observed relationship between the two quantities. In the revised text, we explain that under the assumption of $E_a = f(E_p/P)$, with f being any function, the effects of changes in $E_p$ or in P are expected to be the same as long as the ratio, $E_p/P$, remains constant (which is the kind of changes we considered in our*

45 *paper). We believe that the observed difference in the responses to changes in $E_p$ and changes in P is due to the fact that the $E_a$ depends on the water content of the top soil layer. The water content of the top soil layer is a function of the specific temporal fluctuations of the precipitation and the potential evapotranspiration as well as of the soil hydraulic parameters.*
*We revised the paragraph to reflect the changes outlined above.*

**General comment 4:** The model assumes that all evaporation is soil evaporation and no overland flow. It is unclear why this
50 is realistic even with somewhat sparse vegetation. Most of these regions will still have vegetation that evaporates relevant parts of total ET.

**Reply to general comment 4:** *We find that using the global soil parameters and the CRU TS climate data for the locations with sparse vegetation, results in similar simulated groundwater recharge and independently reported groundwater recharge. Therefore, it is justified to neglect the specific details of vegetation transpiration (information that we do not have for all the*
55 *locations considered) within the scope of our study.*

**General comment 5:** The manuscript states surprise that the extreme ETref statistics have much weaker effects on the GR than changes in extreme rain statistics. Isn't this a result in line with obvious expectation?

**Reply to general comment 5:** *We found that changes in the mean $E_p$ have a greater impact on R/P than changes in the mean precipitation. Therefore, we find it interesting that for changes in the extreme value statistics, the impact of changes in $E_p$ is*
60 *weaker than the impact of changes in P.*

**General comment 6:** The use of symbols is highly confusing with GR standing for "recharge" and R for "rain". However, to readers it would make interpretation a lot easier if a single letter was used for a process, and maybe a subscript is needed for further specifications. This avoids that GR reads as G times R. In addition, the use of P for precipitation, and R for recharge seems slightly more conventional. Such formulation would also come in handy when the symbols are used to derive new
65 equations (Eq. 6-9).

**Reply to general comment 6:** *We accept the comment.*
*We changed the notation. In the revised MS, R represents the groundwater recharge rate, P represents the precipitation rate, $E_p$ represents the potential evapotranspiration and $E_a$ represents the actual evapotranspiration.*

**70 References**

Taylor, R. G., Todd, M. C., Kongola, L., Maurice, L., Nahozya, E., Sanga, H., and MacDonald, A. M.: Evidence of the dependence of groundwater resources on extreme rainfall in East Africa, Nature Climate Change, 3, 374–378, https://doi.org/10.1038/nclimate1731, 2013.

---

## Author Comment (AC2)

**Response to comments made by Reviewer 2 on a manuscript entitled 'Changes in mean evapotranspiration dominate groundwater recharge in semi-arid regions'**

Tuvia Turkeltaub[1] and Golan Bel[2,3]

[1]Zuckerberg Institute for Water Research, Blaustein Institutes for Desert Research, Ben-Gurion University of the Negev, Sede Boqer Campus 8499000, Israel
[2]Department of Environmental Physics, Blaustein Institutes for Desert Research, Ben-Gurion University of the Negev, Sede Boqer Campus 8499000, Israel
[3]Faculty of Environmental Sciences, Czech University of Life Sciences Prague, Kamýcká 129, Praha – Suchdol, 165 00, Czechia

**Correspondence:** Tuvia Turkeltaub (tuviat@bgu.ac.il)

**Reviewer 2**

**General comments**

**General comment 1:** Dear Editor, I have carefully read the submitted manuscript, which aims to evaluate the impact of the variation in mean and standard deviation of the statistical distributions of rainfall (R) and potential evapotranspiration (ETref) on groundwater recharge (GR). I am in trouble, because the objective of the study seemed quite interesting and totally in line with the HESS journal topics, but I note that behind the desire to internationalize the study with global datasets (a condition rightly much appreciated by the journal) there are major uncertainties on the definition of the parameters (GR).

**Reply to general comment 1:** *Our precise definition of groundwater recharge is specified in the second sentence of the introduction. Due to the lack of information regarding the variability of the water table depth in all the locations considered, we are limited in considering the amount of water infiltrating deep enough so that it is not lost to evaporation, transpiration, or runoff. The depth of 5m that we considered satisfies these requirements, given the sparse vegetation in those locations. The main difference between our definition and the definition suggested by the reviewer is the travel time. The travel time differences can induce a lag between the infiltration to the depth we considered and the actual depth of the water table. However, for the long-term simulations we considered, and given the 200 years of spin-up we used to ensure independence from the initial conditions, the accumulated infiltration flux is not expected to change much by the different travel times.*

We revised the second sentence of the introduction as follows: R is the amount of water infiltrating the soil deep enough such that it is not lost to evaporation, transpiration, or runoff. Note that this definition is not the same as the definition of some authors, which define it as the amount of water replenishing the aquifer (Healy and Scanlon, 2010) (the main difference is the travel time)(lines 11-15).

**General comment 2:** on the relevance of the simulations to the reality of the physical data (ETref and R variability)

**Reply to general comment 2:** *Common climate projections suggest that changes to the potential ET are mostly limited to the range of 10% Basso et al. (2021); Guo et al. (2023). Changes to the precipitations are much larger. To illustrate the latter we present in the figure below the predicted changes for a location in Australia. The different models suggest different changes spanning the range we considered. From a theoretical point of view, it is interesting to study the effects of small and large changes in climate conditions as we did.*

[Figure]

**Figure 1.** Probability density functions (PDF) of the yearly rainfall (mm/year) at three selected locations in Australia. The PDF of the CRU data represents the yearly rain statistics from 1958 to 2014. The yearly rain PDFs of four CIMP6 models (*cesm2*, *cmcc_cm2_sr5*, *fio_esm_2_0*, *noresm2_mm*) are accompanied with the CRU data to display future projections for yearly rain (2050-2099).

**General comment 3:** and on the initial hypotheses (infiltration and soil heterogeneity considered only for the first cm of topsoil).

**Reply to general comment 3:** *We considered the top heterogeneity to evaluate its effect on evaporation, runoff, and deep drainage. We found that there were no significant changes in deep drainage rate (our definition of groundwater recharge) when accounting for the top soil heterogeneity (relative to considering a uniform column with the hydraulic parameters of the top (0–5cm) soil layer). This is valid for the many semi-arid locations we considered where the travel time (to the depth where the infiltration rate was recorded in the simulations) is much shorter than the duration of the simulation.*

*We added a reference to the work of Or et al. (2013) which supports our assumption (line 98)*

**Reply to general comment 4:** I don't want to sound too critical, but I think there are some gaps in this study, which need to be fixed at least by defining the limits more and also recalibrating the text and parameter definition. For example, Regarding

the GR parameter there is a big divergence between what I mean (and also other authors) and what I read here. I suggest the authors to keep this reference in mind: https://doi.org/10.1017/CBO9780511780745.

**Reply to general comment 4:** *The R was defined in the original text. Following the comment, we added an explanation that this definition differs from the definition used by some other authors. We also added a citation of the book listed by the reviewer.* *We added the following text: 'Note that this definition is not the same as the definition of some authors, which define it as the amount of water replenishing the aquifer (Healy and Scanlon, 2010) (the main difference is the travel time).'(lines 11-15).*

**General comment 5:** The last analytical part which considers the derivative of the groundwater recharge (GR) and Rainfall (R) ratio (GR/R) with respect to ET is not correct, in my opinion. Actual evapotranspiration depends both on rainfall and potential evapotranspiration and cannot be considered as independent. Based on that part, conclusion section could be misleading too.

**Reply to general comment 5:** *Indeed, we find that in many locations there is no linear relationship between the actual evapotranspiration ($E_a$) and the potential evapotranspiration ($E_p$). In the revised text, we explain that under the assumption of $E_a = f(E_p/P)$, with f being any function, the effects of changes in $E_p$ or in P are expected to be the same as long as the ratio, $E_p/P$, remains constant (which is the kind of changes we considered in our paper). We believe that the observed difference in the responses to changes in $E_p$ and changes in P is due to the fact that the $E_a$ depends on the water content of the top soil layer. The water content of the top soil layer is a function of the specific temporal fluctuations of the precipitation and the potential evapotranspiration as well as of the soil hydraulic parameters.*
*We revised the paragraph to reflect the changes outlined above (lines 238-242).*

**General comment 6:** I would like the authors to explain more in depth the differences between this new study and the previous one (also published on HESS, https://doi.org/10.5194/hess-27-289-2023).

**Reply to general comment 6:** *Our previous paper that was published in HESS* **'The effects of rain and evapotranspiration statistics on groundwater recharge estimations for semi-arid environments'** *addressed the issue of correct representation of observation data in synthesized stochastic time series of rain and potential evapotranspiration. It presented a novel method to generate daily precipitation (P) and $E_p$ time series according to the monthly empirical distributions of climate records. The observation data includes many different aspects of the statistics, and when a synthetic series is generated, some of the characteristics are often lost. Therefore, we examined the sensitivity of R fluxes calculations to the correction methods and pointed out the best method for adequate simulations of groundwater recharge.*
*In this paper, we address a totally different question. We analyze the effects of changes in the statistics of $E_p$ and P on R. The method suggested in the previous paper is used in the current paper to generate the stochastic series, and then the effects of changes in the mean, standard deviation, and frequency of extreme events on R are examined. To the best of our knowledge, this question has not been addressed in previous studies.*
*A few studies focused on groundwater recharge under projected climate conditions (based on specific GCM simulations) and reached contradicting conclusions regarding future groundwater recharge. Our study explains the effects of different changes*

*to the statistics and, thereby, the origin of different outcomes while using different GCM simulations.*

*In our current study, we attempted to address this complexity by separately considering different changes in climate statistics. Namely, we consider separate changes in the mean, standard deviation (STD), and extreme events of $E_p$ and P statistics (relative to the measured statistics) in multiple locations across the globe. We demonstrate that the effect of changes in the average $E_p$ on R is substantially greater than changes in the average P and changes in the STD of the $E_p$ and the P. Changes in the statistics of extreme rainfall events have a much greater effect than extreme $E_p$ events on R. In general, we found nonlinear responses of the R to changes in the climate variable statistics. We suggest that the differences in climate statistics may explain the contradictory results of previous R studies.*

**General comment 7:** It would be desirable for the authors to make the raw datasets and part of the simulations available and public as supplementary material. Surely, it would also be helpful for us reviewers to confirm or resolve some doubts.

**Reply to general comment 7:** *All data is available in Hydroshare at the following link: http://www.hydroshare.org/resource/ 5a28b74c55c74a7e9ad2b59a0a5d9ab3*

**Specific comments**

**Specific comment 1:** Line 3: indices, parameters or "characteristics" of the statistical distribution?

**Reply to specific comment 1:** *We mean characteristics, just as written. The characteristics include changes to the mean, variance, frequency of extreme events, and other changes. It is not a simple change in parameters of a known analytical form of the $E_p$ distribution.*

**Specific comment 2:** Line 3: Why do you rename Potential Evapotranspiration as ETref? Usually it is widely accepted the abbreviation PET. I suggest you to use it to be clearly understood by all the readers. ETref is usually the reference EvapoTranspiration, related to specific crop, isn't it? Could you explain better the differences (maybe in the introduction section)?

**Reply to specific comment 2:** *We accept the comment and changed the notation.*
*In the revised version $E_p$ represents the potential evapotranspiration. $E_p$ is explicitly specified in section 2.2, "The daily $E_p$ values were calculated according to the Penman-Monteith method using climate variables such as mean, maximum, and minimum temperatures, vapor pressure, cloud cover, and wind speed (Harris et al., 2014)."*

**Specific comment 3:** Line 5: standard deviation (STD). The first time you present an abbreviation, even if widely known, please always report the extended word before.

**Reply to specific comment 3:** *We accept the comment and changed the text accordingly (Line XXX).*

**Specific comment 4:** Lines 7-8: Actually, this seems to be normal, isn't it? The extreme statistics is mainly used for rainfall variability analyses.

**Reply to specific comment 4:** *We found that changes in the mean $E_p$ have a greater impact on R/P than changes in the mean precipitation. Therefore, we find it interesting that for changes in the extreme value statistics, the impact of changes in $E_p$ is weaker than the impact of changes in P.*

**Specific comment 5:** Lines 11-12: some suggestions on other recent studies focusing on groundwater recharge and related water management issues:

In particular:

A recent book describing methods for assessing groundwater recharge (physical methods, inverse methods, lumped models, water budgets etc..) https://doi.org/10.1017/CBO9780511780745

Recent studies based on GR estimation using different methods:

https://doi.org/10.5194/hess-25-2931-2021

https://www.sciencedirect.com/science/article/pii/S2352801X18302662

https://link.springer.com/article/10.1007/s40710-024-00688-5

https://www.sciencedirect.com/science/article/pii/S0022169417307436

https://www.sciencedirect.com/science/article/pii/S2214581821001464

**Reply to specific comment 5:** *We thank the reviewer for specifying these references.*

*The references are cited in the revised manuscript (lines 12-13).*

**Specific comment 6:** Lines 12-13: "Groundwater recharge is defined in a general sense as the volume or process of downward flow of water reaching the water table, forming an addition to the groundwater reservoir" (Hartmann A., The Hydrology of Groundwater Systems - From Recharge to Discharge, in Encyclopedia of Inland Waters (Second Edition), 2022)

I think this definition is more correct. In my opinion, one of the potential weaknesses of this study is assessing the GR without adequately considering the geological characteristic of the unsaturated zone in the study areas. GR is the water volume/flow that actually reaches the water table. Is your control volume limited to the topsoil? What do you mean with "deep enough"?

**Reply to specific comment 6:** *Please see our response to the reviewer's general comment 4. By "deep enough," we mean a depth from which there is no water loss for ET. Specifically, in this study, this depth was set to 5m below the surface. The difference between our definition and other definitions was specifically mentioned at the beginning of the Introduction section.*

**Specific comment 7:** Lines 16: Other suggestions on references:

https://www.sciencedirect.com/science/article/pii/S0013935123023368

https://doi.org/10.3390/w15142610

https://doi.org/10.7343/as-2019-373

https://link.springer.com/article/10.1007/s10113-023-02114-2

**Reply to specific comment 7:** *We added the suggested references to the revised manuscript (lines 19-20).*

**Specific comment 8:** Line 34: 'potential ET (ETref)'

**Reply to specific comment 8:** *See our response to the reviewer's specific comment 2. ETref was replaced by $E_p$*

**Specific comment 9:** Lines 42-43: I also read your previous paper (https://doi.org/10.5194/hess-27-289-2023, 2023.). Can you explain what novelty does this new study bring? What are the main differences?

**Reply to specific comment 9:** *Please see our response to the reviewer's general comment 6. For convenience, we respond here again. Our previous paper that was published in HESS* **'The effects of rain and evapotranspiration statistics on groundwater recharge estimations for semi-arid environments'** *addressed the issue of correct representation of observation data in synthesized stochastic time series of rain and potential evapotranspiration. It presented a novel method to generate daily precipitation (P) and $E_p$ time series according to the monthly empirical distributions of climate records. The observation data includes many different aspects of the statistics, and when a synthetic series is generated, some of the characteristics are often lost. Therefore, we examined the sensitivity of R fluxes calculations to the correction methods and pointed out the best method for adequate simulations of groundwater recharge.*

*In this paper, we address a totally different question. We analyze the effects of changes in the statistics of $E_p$ and P on R. The method suggested in the previous paper is used in the current paper to generate the stochastic series, and then the effects of changes in the mean, standard deviation, and frequency of extreme events on R are examined. To the best of our knowledge, this question has not been addressed in previous studies.*

*A few studies focused on groundwater recharge under projected climate conditions (based on specific GCM simulations) and reached contradicting conclusions regarding future groundwater recharge. Our study explains the effects of different changes to the statistics and, thereby, the origin of different outcomes while using different GCM simulations.*

*In our current study, we attempted to address this complexity by separately considering different changes in climate statistics. Namely, we consider separate changes in the mean, standard deviation (STD), and extreme events of $E_p$ and P statistics (relative to the measured statistics) in multiple locations across the globe. We demonstrate that the effect of changes in the average $E_p$ on R is substantially greater than changes in the average P and changes in the STD of the $E_p$ and the P. Changes in the statistics of extreme rainfall events have a much greater effect than extreme $E_p$ events on R. In general, we found nonlinear responses of the R to changes in the climate variable statistics. We suggest that the differences in climate statistics may explain the contradictory results of previous R studies.*

**Specific comment 10:** Line 44: '... assuming that this flux reaches the water table' In my opinion, this could be a too much strong condition, difficult to be respected in some specific hydrogeological frameworks.

**Reply to specific comment 10:** *For the hydrological framework considered in this manuscript, this assumption is more than reasonable because we found no evidence for water loss to the atmosphere from depths of a few tens of cm and deeper.*

**Specific comment 11:** Lines 57-58: why? I mean, did you choose them specifically?.

**Reply to specific comment 11:** *We did choose the locations specifically. First, we only considered locations with semi-arid or arid conditions. Secondly, to avoid the need for specific vegetation water uptake characteristics that we do not have we focus on locations where the water uptake by vegetation may be neglected.*

**Specific comment 12:** Line 84: again, this is another very strong assumption not applicable to many contexts

**Reply to specific comment 12:** *Please see our response to the reviewer's specific comment and general comments regarding our definition of R (previously GR). We found no evidence of water loss from depths greater than 30cm (Or et al., 2013). We added this reference to the revised text*

**Specific comment 13:** Line 85: I take a look at SI. The unit measure of each parameter is missing. Please add it.

**Reply to specific comment 13:** *We thank the reviewer for the comment.*
*The units were added to the datasets now available in the following link:*
*http://www.hydroshare.org/resource/5a28b74c55c74a7e9ad2b59a0a5d9ab3.*

**Specific comment 14:** Lines 96-97: what does it mean observed GR/Rain ratio. Do you measure GR?

**Reply to specific comment 14:** *For clarity, we changed the wording from observed to reported. As mentioned in the manuscript, all reported values were estimated using ground-based methods such as chloride mass balance and water isotopes. See the beginning of section 2.1*

**Specific comment 15:** Lines 114-120: I think all this procedure is important and should be described better. Some sentences are misleading. Authors can also take into account explaining the procedures for the mean and st.dev modification with a flowchart to be added in this part of the text (i.e. a figure).

**Reply to specific comment 15:** *We thank the reviewer for the comment.*
*As suggested, a flowchart was created and added to the supporting information.*

**Specific comment 16:** caption of Figure 1: I still don't understand properly what you mean with measured GR. Is the measured value coming from the water content/flux detected by satellite observation at 5 m depth? Moreover, you previosly affirmed (pag 3) that for the water flow simulation you consider only three layers of the topsoil up to 30 cm. Please try to clearly define limits and boundaries for both measured and calculated GR (what about a table?)

**Reply to specific comment 16:** *See reply to specific comment 14, we have changed the wording from 'measured' to 'reported'.*

**Specific comment 17:** Line 134: This second part is clearer because of the presence of different formulas defining MTS. Anyway I still suggest to consider a flowchart to let the reader better understand the procedure.

**Reply to specific comment 17:** *See our reply to specific comment 15.*

**Specific comment 18:** Line 161-162: Really? How much does the average annual temperature decrease to have a 33% decrease in evapotranspiration?

**Reply to specific comment 18:** *See the response to general comment 2. Common climate projections suggest that changes to the potential ET are mostly limited to the range of 10% Basso et al. (2021); Guo et al. (2023). Changes to the precipitations are much larger. To illustrate the latter we present in the figure below the predicted changes for a location in Australia. The different models suggest different changes spanning the range we considered. From a theoretical point of view, it is interesting to study the effects of small and large changes in climate conditions as we did.*

**Specific comment 19:** Lines 189-190: what about the rainfall intensity? If you consider extreme events you should take into account it, keeping in mind that an increase in intensity also means a lower capacity of the soil to let water infiltrate

**Reply to specific comment 19:** *In this work, we considered daily values of rain and $E_p$; obviously, when the daily rain amount changes, so does the intensity. Characteristics of the daily distribution were not considered in the current study.*

**Specific comment 20:** Lines 212-223:I don't appreciate this part. I find these considerations extremely forced. Where are the soil properties in here? Moreover, this is an analytical guise which, however, is not adherent to the reality of measured data in the field. And this is because the initial definition is, in itself, incorrect. Groundwater recharge is not what you define in (6), where with P-ET you are representing the net or effective rainfall, i.e. the quantity of water available for infiltration, net of evaporation and transpiration processes that occur. After this, soil and rock mass properties have key role in defining runoff and infiltration. I understand and appreciate the effort and will to try to give a global picture of such an important process for water management, but at this point in the manuscript, I am convinced that there are major gaps from a conceptual point of view in this study. The hydrogeological characteristics (larger orders of magnitude, not only the topsoil in the first meter) of the outcropping formations in the locations are not considered. Rainfall and evapotranspiration are considered to have potentially similar variability in your simulations. However, average rainfall, within its seasonality, clearly has greater variability on a monthly and annual scale when intense drought or storm events occur. On the contrary, evapotranspiration can certainly have a growing role considering the average increase in global temperatures, but it always depends on a parameter (T) that has a lower variability and this can be found in the historical series of any air thermometer in the world.

**Reply to specific comment 20:** *See reply to general comment 5. We found that in many locations, there is no linear relationship between the actual evapotranspiration ($E_a$) and the potential evapotranspiration ($E_p$). In the revised text, we explain that under the assumption of $E_a = f(E_p/P)$, with $f$ being any function, the effects of changes in $E_p$ or in $P$ are expected to be the same as long as the ratio, $E_p/P$, remains constant. We believe that the observed difference in the responses to changes in $E_p$ and changes in $P$ is due to the fact that the $E_a$ depends on the water content of the top soil layer. The water content of the top soil layer is a function of the specific temporal fluctuations of the precipitation and the potential evapotranspiration as well as of the soil hydraulic parameters.*

**Specific comment 21:** Lines 222-223: In the last sentence, assuming that ET is proportional to ETref is misleading too, considering what you state after. The subsequent implication is then referred to rainfall (weaker sensitivity of GR to R changes). Actually ET depends both on ETref and R! If ETref > R, the ET = R. So, at the time scale considered, the R acts as a cutoff for ET if ETref is higher than R (nothing can evapotranspirate if there is no rainfall). For this reason I cannot accept this analytical part and I think it's better to delete it. Because actually ET is a function of R too (ET=ET(R)) and not a indipendent variable.

Just to give an exemple: annual estimation of ET by Turc empirical formula (the easiest way to find it), needs the cumulative annual rainfall to be considered as input.

**Reply to specific comment 21:** *See the response to the comment above and to general comment 5.*

**Specific comment 22:** Lines 236-238: This distinction is very important and probably should have been stated much earlier, not at the end of the manuscript. Karst settings are charcterized by focused GR and host some of the most important groundwater resources in the world.

**Reply to specific comment 22:** *We accept the comment and added a sentence emphasizing this distinction in the last part of the introduction.*
*In areas where R occurs predominantly through focused processes, additional factors dominate the overall recharge and are beyond the scope of this study (lines 49-50).*

**Specific comment 23:** Line 242: I would like to receive your datasets, if it is possible. Thank you.

**Reply to specific comment 23:** *See reply to general comment 7. All the data related to this work is uploaded to the HydroShare and is available at the following link:http://www.hydroshare.org/resource/5a28b74c55c74a7e9ad2b59a0a5d9ab3.*

**References**

Basso, B., Martinez-Feria, R. A., Rill, L., and Ritchie, J. T.: Contrasting long-term temperature trends reveal minor changes in projected potential evapotranspiration in the US Midwest, Nature Communications, 12, 1476, 2021.

Guo, D., Olesen, J. E., Manevski, K., Pullens, J. W. M., Li, A., and Liu, E.: Change Trend and Attribution Analysis of Reference Evapotranspiration under Climate Change in the Northern China, Agronomy, 13, https://doi.org/10.3390/agronomy13123036, 2023.

Healy, R. W. and Scanlon, B. R.: Estimating Groundwater Recharge, Cambridge University Press, 2010.

Or, D., Lehmann, P., Shahraeeni, E., and Shokri, N.: Advances in soil evaporation physics—A review, Vadose Zone Journal, 12, 1–16, 2013.

---

## Author Comment (AC3)

**Response to comments made by a member of the scientific community 1 on a manuscript entitled 'Changes in mean evapotranspiration dominate groundwater recharge in semi-arid regions'**

Tuvia Turkeltaub[1] and Golan Bel[2,3]

[1]Zuckerberg Institute for Water Research, Blaustein Institutes for Desert Research, Ben-Gurion University of the Negev, Sede Boqer Campus 8499000, Israel
[2]Department of Environmental Physics, Blaustein Institutes for Desert Research, Ben-Gurion University of the Negev, Sede Boqer Campus 8499000, Israel
[3]Faculty of Environmental Sciences, Czech University of Life Sciences Prague, Kamýcká 129, Praha – Suchdol, 165 00, Czechia

**Correspondence:** Tuvia Turkeltaub (tuviat@bgu.ac.il)

**Community comment 1**

**General comment:** Good research in the field of groundwater hydrology that has been approached with a worldwide angle. However, important detail is missing. Please, take into account my minor points to fix the issues.

**Reply to general comment:** *Thank you for acknowledging the contribution of our work to the field of groundwater hydrology. In what follows we address your helpful specific comments.*

**Specific comments**

**Specific comment 1:** Line 18. "In recent years, much effort has been devoted to the analysis of the sensitivity of groundwater systems to climate change". Add recent literature on the effects of climate change in mountain ranges, the aquifer recharge from the snow is very sensitive to the climate:

- Lorenzi, V., Banzato, F., Barberio, M. D., Goeppert, N., Goldscheider, N., Gori, F., Lacchini A., Manetta M, Medici G, Rusi S, Petitta, M. (2024). Tracking flowpaths in a complex karst system through tracer test and hydrogeochemical monitoring: Implications for groundwater protection (Gran Sasso, Italy). Heliyon, 10(2).

- Langman, J. B., Martin, J., Gaddy, E., Boll, J., & Behrens, D. (2022). Snowpack aging, water isotope evolution, and runoff isotope signals, Palouse Range, Idaho, USA. Hydrology, 9(6), 94.

**Reply to specific comment 1:** *Thank you for bringing those works to our attention.* *We added the suggested references to the revised manuscript (line 26).*

**Specific comment 2:** Line 48. Cleary mention the 3 to 4 specific objectives of your research by using numbers (e.g., i, ii and iii).

**Reply to specific comment 2:** *We accept the comment and have numbered the objectives of our research (line 52).*

**Specific comment 3:** Line 78. "l is the pore connectivity". Your research appears to focus on porous aquifers of siliciclastic nature (plio-quaternary age?). This point is not clear by reading the manuscript.

**Reply to specific comment 3:** *We accept the comment and revised the text to indicate that in our study, the unsaturated zone is considered to consist of siliciclastic materials (line 92).*

**Specific comment 4:** Lines 80-81."Sand, silt, and clay contents". The geological nature of your aquifers have not been disclosed, see also my comment above.

**Reply to specific comment 4:** *See our reply to specific comment 3.*

**Specific comment 5:** Line 184. "Under some future climate predictions, the frequency of extreme events is expected to double". Please, be more specific. Are you talking about semi-arid / arid regions?

**Reply to specific comment 5:** *Myhre et al. (2019) do not specify changes in particular regions but rather illustrate projected changes in extreme rainfall statistics in Europe and the United States. We investigated the potential impact of these changes in extreme statistics on groundwater recharge in semi-arid and arid areas. The goal of our study is not to quantify the effects of specific predicted future climate but rather to quantify the effects of various changes in the climate statistics on groundwater recharge.*

**Specific comment 6:** Line 184. "Under some future climate predictions, the frequency of extreme events is expected to double". This sentence should be expanded and moved to the discussion section.

**Reply to specific comment 6:** *The changes in extreme rainfall statistics was reported by Myhre et al. (2019) and not in the current study. Note that we only examined the impact of this possible change in extreme climate statistics on groundwater recharge.*

**Specific comment 7:** Lines 231-240. The conclusion is too short, it needs more detail.

**Reply to specific comment 7:** *We accept the comment and have expanded the text in Section 5 (Conclusions) to briefly explain our main goal and describe the conclusions drawn in our study. (lines 250-280).*

**Specific comment 8:** Lines 232-234. "Our results suggest...rainfall statistics". The sentence is unclear and too long. Please, split it in two parts.

**Reply to specific comment 8:** *We accept the comment and revised the text accordingly (lines 263-267).*

**Specific comment 9:** Line 237. "Focused processes". Which processes? Please, be more specific.

**Reply to specific comment 9:** *We mainly refer to preferential flow and recharge of runoff at specific locations on the landscape. We added the above definition to the text (lines 276-277).*

**Specific comment 10:** Figure 1a. You also have study sites and aquifers in highly arid settings, this is not clear in the text. You don't have only semi-aridity.

**Reply to specific comment 10:** *We revised the text to indicate that arid regions are also considered.*

**Specific comment 11:** Figure 1c. You can also report Mean Error, Mean Absolute Error and RMS in the graph.

**Reply to specific comment 11:** *We provided the suggested statistical indices for model performance evaluation in the caption of Figure 1.*

**References**

Myhre, G., Alterskjær, K., Stjern, C. W., Hodnebrog, Ø., Marelle, L., Samset, B. H., Sillmann, J., Schaller, N., Fischer, E., Schulz, M., et al.: Frequency of extreme precipitation increases extensively with event rareness under global warming, Scientific reports, 9, 16 063, https://doi.org/10.1038/s41598-019-52277-4, 2019.

---

## Author Comment (AC4)

**Response to comments made by a member of the scientific community 2 on a manuscript entitled 'Changes in mean evapotranspiration dominate groundwater recharge in semi-arid regions'**

Tuvia Turkeltaub[1] and Golan Bel[2,3]

[1]Zuckerberg Institute for Water Research, Blaustein Institutes for Desert Research, Ben-Gurion University of the Negev, Sede Boqer Campus 8499000, Israel
[2]Department of Environmental Physics, Blaustein Institutes for Desert Research, Ben-Gurion University of the Negev, Sede Boqer Campus 8499000, Israel
[3]Faculty of Environmental Sciences, Czech University of Life Sciences Prague, Kamýcká 129, Praha – Suchdol, 165 00, Czechia

**Correspondence:** Tuvia Turkeltaub (tuviat@bgu.ac.il)

**Community comment 2**

**General comment 1:** The present paper addresses the impact of changes in rainfall distribution and potential evapotranspiration on groundwater recharge (GR) in semi-arid areas. In their introduction based on the scientific literature, the authors note that "no conclusive generic outcomes can be drawn regarding the relationship between changes in climate conditions and the resulting changes in GR rates", "it is unclear whether the climate variability is amplified or smoothed in the GR response" and "even the trend of the GR response is uncertain". These assessments are reasonable and I fully agree with them.

**Reply to general comment 1:** *These assessments were drawn from previous studies, and those are cited in the paper.*

**General comment 2:** Several factors may explain this large range of GR responses to present climatic changes, which leads to apparent contradictions in their recent evolution. The first explanation is the variability of environmental conditions in the natural state. The second explanation is the multiplicity of scientific approaches (various methods using various types of data sets monitored at different scales in space and over time), which logically lead to heterogenous results. Moreover, depending on methods, the calculated GR represents an integration over a very variable time. The third explanation is that in the last decades GR may have changed a lot as a consequence of the climate change and the multifaceted human modifications of semi-arid landscapes (e.g. changes in land use and land cover; water conservation works). The GR estimates found in the literature aggregate values from various stages of this evolution between areas still in a mostly natural state and others deeply modified. Depending on areas, direct human intervention may be a much stronger factor than climate change. For instance, the increase in GR by one order of magnitude in southwestern Niger (e.g. Favreau et al., 2009) and by two orders of magnitude in eastern Australia (e.g. Allison et al., 1990) was explained by a change in the vegetation cover only. Therefore raw data from

the literature should be used with a cautious reference to their specific contexts, which is not really the case in the submitted text.

**Reply to general comment 2:** *We agree with the reviewer's perspective regarding the large uncertainty in the reported groundwater recharge fluxes (see reply to general comment 2 of reviewer #1). We believe that the explanations provided by the reviewer can clarify why our model did not adequately fit some of the reported groundwater recharge fluxes. This description has been added to the revised text, along with relevant references.*

*We revised the text to include the explanations above (lines 67-72).*

**General comment 3:** An important concern is the geographical extent of this work. The authors used GR estimates from 200 semi-arid locations in different continents, in a wide range of soils and climate conditions. The annual rainfall in the sites considered ranges from 180 to 1044 mm, with more than one half between 400 and 600 mm. In fact, 60 % of the sites are in Australia, 20 % in Africa, 10 % in North America and 10 % in the other continents, which differs significantly from the distribution of semi-arid areas in the world. Does this selection bias the final conclusions? The Figure 1-a will probably surprise many readers who usually see a much larger extent of semi-arid areas in global maps; this singularly restricted coverage should be justified.

**Reply to general comment 3:** *We compare the cumulative distribution of results from 67 randomly sampled locations in Australia with the 67 locations outside Australia in the graph below. The comparison clearly shows that changes to the mean $E_p$ would considerably change the R/P ratio in Australia and other arid and semi-arid locations. This result supports our interpretation that the results are not limited to the region of Australia.*

[Figure]

**Figure 1.** Cumulative distributions of the change in the R/P ratio, $\Delta(R/P)$, for 67 random locations in Australia (left panel) and 67 locations in other continents (right panel). The different lines indicate different changes in the mean $E_p$.

**General comment 4:** Another important concern is the very restrictive assumptions for the calculation: (i) GR occurs only through diffuse recharge (i.e. without any focused recharge); (ii) transpiration is negligible vs. evaporation; (iii) surface runoff is negligible; (iv) there is no preferential flow in the unsaturated zone. The fist assumption contradicts the observation that focused and diffuse recharge often coexist in the same area. Their respective proportions depend on local geomorphological conditions (e.g. Cuthbert at al., 2019). At the global scale, it is generally accepted that the proportion of focused recharge increases with aridity and as a consequence the driest semi-arid regions would be excluded from this calculation. The second assumption neglects transpiration uptake while the vegetation cycle in semi-arid areas is closely linked with the rain distribution, which is also the driver for GR. The third assumption requires to limit the application of the calculation to very flat areas and/or very low rainfall. The fourth assumption requires a very poor biological activity (roots and fauna).

These four assumptions together are so constraining that the geographical extent of the concerned semi-arid areas is probably very small. The practical relevance of this text appears therefore limited and the added value for researchers working on groundwater in semi-arid areas may be seriously questioned. The authors are conscious of these weaknesses and in their conclusion they mention the possibility of extending their work, but this last precaution is not enough to give the text a convincing strength.

**Reply to general comment 4:** *We conducted our analysis using locations where ground-based methods are employed to estimate groundwater recharge. Methods like chloride mass balance (CMB) and water isotopes are typically used to assess diffuse recharge. While these methods can indicate focused recharge, they may not efficiently quantify the focused recharge component. Additionally, we did not ignore transpiration; rather, we excluded locations with significant plant cover to reduce uncertainty and focus on illustrating the effects of changes in precipitation and potential evapotranspiration on groundwater recharge. Previous paper presented contradicting results regarding the effect of future projections on groundwater recharge. n our study, we propose potential sources of uncertainty and recommend that future studies analyze climate statistics initially to better explain projected changes.*

---

## Author Response (AR2)

**Response to referees' comments on manuscript entitled 'Changes in mean evapotranspiration dominate groundwater recharge in semi-arid regions'**

Tuvia Turkeltaub[1] and Golan Bel[2,3]

[1]Zuckerberg Institute for Water Research, Blaustein Institutes for Desert Research, Ben-Gurion University of the Negev, Sede Boqer Campus 8499000, Israel

[2]Department of Environmental Physics, Blaustein Institutes for Desert Research, Ben-Gurion University of the Negev, Sede Boqer Campus 8499000, Israel

[3]Faculty of Environmental Sciences, Czech University of Life Sciences Prague, Kamýcká 129, Praha – Suchdol, 165 00, Czechia

**Correspondence:** Tuvia Turkeltaub (tuviat@bgu.ac.il)

Dear Editor,

For clarity, we use regular black font for the quoted reviewer comments and *blue* italicized font for our responses. The line numbers specified in our responses refer to the track changes pdf attached.

**Editor**

**Comment 1:** Dear authors, thanks for the resubmission. As you can see, both referees agree the revisions have improved the paper. I agree. One referee is satisfied, the author not that enthusiastic. From my own reading I think a bit more discussion on the limitations of the selected model and how this could be 'meaningfully' used in climate sensitivity analysis is required. Also, quite some studies show that diffusive flow in the unsat zone in semi-arid regions (especially when it is dry) is not the most important process (but pref flow is). Therefore, the applicability of Richards' en MvG equations is a point of concern that would be nice to see in the discussion.

**Reply to comment 1:** *We have expanded the discussion section in the manuscript and elaborated on the limitations of using Richards' equation for estimating the groundwater recharge in locations where considerable runoff is expected (e.g., areas with significant topography and rocky terrains) and where focused recharge is dominant due preferential flow paths (lines 207-216).*

**Comment 2:** The second referee also challenges you to think of the wording as you look at R/P ratios instead of R-values. The referee has a point and I leave it up to you to see how that could be implemented.

**Reply to comment 2:** *We checked again the text in the MS to make sure that there are no misleading wordings. We found that the abstract, indeed, was not clear enough regarding the fact that we focused on the ratio R/P. Therefore, we revised the*

*abstract to reflect that (lines 6-8; 10-11). However, it is important to note that there is only one type of change in the climate conditions, considered in our work, under which the mean precipitation changes. In all other changes, the mean P does not change and changes in R/P are only due to changes in R.*

**Reviewer 1**

**Comment:** I appreciated so much the efforts for improving the overall quality of the paper that is now more acceptable for publication, in my opinion. I think that the limits of the study are now better defined, as well as the focus on evapotranspiration change impacts on groundwater recharge. I'm still not completely convinced about the last analytical section (Section 4), even I acknowledge that it is now better explained with all the hypothesis discussed for the subsequent presentation of formulas. I think that neglecting the runoff could be not applicable in so many contexts (even if semi-arid regions) and that's my only one concern.

**Reply to comment:** *We thank the reviewer for the supportive feedback. Regarding the runoff, we limit our analyses to regions where independent estimations of R agreed with our simulated R, suggesting that our method is adequate for these locations. According to these simulations, the runoff was negligible in most locations. We now further clarify this point in the discussion (lines 213-216).*

**Reviewer 2**

**Comment 1:** I appreciate the authors addressed some of the reviewer comments. However, I understand that studying recharge/precipitation can be (for particular purposes) more relevant than studying absolute recharger rates. However, then it is key that the paper also changes the associated language throughout the entire manuscript. For example, the title right now is already misleading.

**Reply to comment 1:** *We checked again the text of the MS to make sure that there are no misleading wordings. We found that the abstract, indeed, was not clear enough regarding the fact that we focused on the ratio R/P. Therefore, we revised the abstract to reflect that (lines 6-8; 10-11). However, it is important to note that there is only one type of change in the climate conditions considered in our work, under which the mean precipitation changes. In all other changes, the mean P does not change, and changes in R/P are only due to changes in R.*

**Comment 2:** The mathematical derivation provides no theory that can meaningfully be used to describe recharge sensitivity to climate. So I struggle what the value of this is.

**Reply to comment 2:**

*In this research, we found that the $R/P$ ratio exhibits higher sensitivity to changes in the mean potential evapotranspiration ($E_p$) than to changes in the mean annual precipitation ($P$). The mathematical expressions are necessary to explain why this is not trivial and why naively one would expect only sensitivity to the ratio $E_p/P$.*